

# Climate trends in northern Ontario and Quebec from borehole temperature profiles

C. Pickler[1], H. Beltrami[2], and J.-C. Mareschal[1]

[1]GEOTOP, Centre de Recherche en Géochimie et en Géodynamique, Université
du Québec à Montréal, Canada
[2]Climate & Atmospheric Sciences Institute and Department of Earth Sciences, St. Francis Xavier University, Antigonish,
Nova Scotia, Canada

*Correspondence to:* H.Beltrami (hugo@stfx.ca)

**Abstract.** The ground surface temperature histories of the past 500 years were reconstructed at 10 sites containing 18 boreholes
in northeastern Canada. The boreholes, between 400 and 800 m deep, are located north of $51^o$N, and west and east of James Bay
in northern Ontario and Quebec. We find that both sides of James Bay have experienced similar ground surface temperature
histories with a warming of ∼1-2 K for the last 150 years, similar to borehole reconstructions for the southern portion of
the Superior Province and in agreement with available proxy data. A cooling period corresponding to the little ice age was
found at only one site. Despite permafrost maps locating the sites in a region of discontinuous permafrost, the ground surface
temperature histories suggest that the entire region is and was free of permafrost for the past 500 years. This could be the
result of air surface temperature interpolation used in permafrost models being unsuitable to represent the spatial variability of
ground temperatures along with an offset between ground and air surface temperatures due to the snow cover.

# 1   Introduction

Earth's subsurface thermal regime is governed by the outflow of heat from the interior and by temporal variations in ground
surface temperature (GST). The heat flux from the interior of the Earth varies on time scales of the order of a few Myears in
active tectonic regions and several 100 Myears in stable continents. It can be considered as steady state relative to the time
scale of climatic surface temperature variations. To determine the heat flow from the Earth's interior, temperature-depth pro-
files are measured in boreholes. In homogeneous rocks with no heat production, the steady state temperature profile linearly
increases with depth. Persistent temporal changes in the ground surface energy balance cause variations of the ground tem-
perature that diffuse downwards and are recorded as temperature anomalies superimposed on the linear steady-state geotherm
(e.g., Hotchkiss and Ingersoll, 1934; Birch, 1948; Beck, 1977). The extent to which the ground surface temperature changes
are recorded is proportional to their duration and amplitude and inversely proportional to the time when they occurred. For
periodic oscillations of the surface temperature, the temperature is propagated downward as a damped wave. The amplitude
of the wave decreases exponentially with depth over a length scale $\delta$ (skin depth) proportional to the square root of the period
($\delta = \sqrt{\kappa T/\pi}$), where $\kappa$ is the thermal diffusivity of the rock, $\approx 10^{-6}$ m$^2$ s$^{-1}$ or $\approx 31.5$ m$^2$ yr$^{-1}$. This damping removes the
high-frequency variability that is present in meteorological records and allows for the preservation of the long-term climatic





trends in the ground temperature signal (e.g., Beltrami and Mareschal, 1995).

From the interpretation of temperature-depth profiles, it is possible to infer centennial trends in Earth's surface temperature variations. The first attempt to infer climate history from temperature-depth profiles was the study by Hotchkiss and Inger-
soll (1934) who estimated the timing of the ice retreat at the end of the last glaciation. It was, however, not until the 1970s that systematic studies were undertaken to infer past climate from such profiles (e.g., Cermak, 1971; Sass et al., 1971; Vasseur et al., 1983). In the 1980s, with increasing concern over global warming, use of borehole temperature-depth profiles to estimate recent (<300 years) climate change became widespread following the study of Lachenbruch and Marshall (1986). This has lead to many local, regional, and global studies (e.g., Huang et al., 2000; Harris and Chapman, 2001; Gosselin and Mareschal,
2003; Beltrami and Bourlon, 2004; Pollack and Smerdon, 2004; Chouinard et al., 2007; Pickler et al., 2016).

Because of the availability of suitable temperature-depth profiles in the Canadian Shield, many studies have been undertaken in central and eastern Canada . The majority of these studies have used temperature-depth profiles from the southern portion of the Superior Province of the Canadian Shield ($\sim$45°-50°N), where many mining exploration holes are readily available and
the crystalline rocks are less likely to be affected by groundwater flow than sedimentary rocks. These studies have shown a warming signal of $\sim$1-2 K over the last $\sim$150-200 years following a period of cooling about 200-500 years BP associated with the little ice age (LIA) (e.g., Beltrami and Mareschal, 1992; Wang et al., 1992; Guillou-Frottier et al., 1998; Gosselin and Mareschal, 2003; Chouinard and Mareschal, 2007) .

For logistical reasons, mining exploration has been restricted to the southernmost part of the Shield and the few holes that have been drilled in northern regions cannot be measured because they are blocked by permafrost. Nevertheless, a few studies were conducted at higher latitudes. Majorowicz et al. (2004) reconstructed the GST history for 61 temperature-depth profiles between 60°N and 82°N in northern Canada. They found strong evidence that GST warming started in the late 18th century and continued until present. Simultaneous inversion of their data yielded a warming of $\sim$2K for the last 500 years. Studies in
Ellesmere Island (above 60°N) have shown varying trends, confirming that temperatures do not increase uniformly over Arctic regions. Taylor et al. (2006) reconstructed the 500 year GST history from three boreholes and found a 3 K warming since the LIA minimum, $\sim$200 yrs BP, which is consistent with Beltrami and Taylor (1995) results and the oxygen isotopes studies on ice cores from the region (Fisher and Koerner, 1994). Chouinard et al. (2007) used three temperature-depth profiles in a region with continuous permafrost at the northernmost tip of Quebec to infer the GST history. They found a very strong and recent
warming of $\sim$2.5 K, with the largest part of this warming occurring in the preceeding 15 yrs, i.e. much later than in Ellesmere Island. Because of lack of adequate borehole temperature depth profiles in eastern Canada between 51°N and 60°N, the large region between the Canadian Arctic and the southern part of the Canadian Shield has not been studied and the climate trends of the last 500 years for this region remain unclear except for boreholes at Voisey Bay, at 56°N on the east coast of Labrador, which show almost no climate signal (Mareschal et al., 2000).




The first motivation of this study is to reduce the gap in data between the Arctic and southeastern Canada. We shall examine 18 temperature-depth profiles measured at 10 sites from eastern Canada to reconstruct the GST histories for the last 500 years. The sites are located in the poorly sampled region north of 51°N, west and east of James Bay in northern Ontario and Quebec. They are to the north of the previous eastern Canada studies and south of the Arctic ones, in a part of the Superior

Province where heat flux is extremely low ($< 30\mathrm{mW\,m^{-2}}$) (Jaupart et al., 2014).

The second motivation of the study is to assess whether borehole temperature profiles can be used to retrace the evolution of permafrost in northern Ontario and Quebec. Permafrost maps locate the Ontario sites in a region classified as with extensive discontinuous permafrost, i.e. where permafrost affects between 50-90% of the ground, while the sites in Quebec are in a

region described as with sporadic discontinuous permafrost, i.e. where less than 50% of the ground is frozen (Brown et al., 2002). In regions with an absence of ground temperature measurements, such as northern Ontario and Quebec, permafrost maps are estimated from surface air temperature and their contour lines (Heginbottom, 2002). The -2.5°C mean annual surface air temperature (SAT) contour line for the period 1950-1980 crosses the southern part of our study region in Ontario, and most of the Quebec sites are located between -2.5 and -5°C SAT contour lines (Phillips, 2002). However, permafrost was not

encountered during sampling of the Quebec or Ontario boreholes. It is also worth pointing out that the ground is covered by thick snow cover during several months (from mid December to late April) in the regions above 50°N. Studies demonstrated that the ground surface temperatures are strongly affected by the duration of the snow cover and are offset from SAT (Bartlett et al., 2005; Zhang, 2005; González-Rouco et al., 2006, 2009; García-García et al., 2016). In these regions with extensive snow cover, the borehole temperature profiles are affected by changes in both SAT and snow cover. Meteorological and proxy data

indicate that there is more snowfall and longer snow cover on the ground in the Quebec region than in Ontario (Bégin, 2000; Brown and Mote, 2009; Brown, 2010; Environment Canada, 2010; Nicault et al., 2014). This points to possibly warmer present ground surface temperatures and smaller permafrost extent in northern Quebec than in Ontario, and the prospect for different ground surface temperature histories between the regions.

## 2   Theory

Assuming Earth is a half-space where physical properties only vary with depth, the temperature at depth $z$, $T(z)$, can be written as (Jaupart and Mareschal, 2011):

$$T(z) = T_o + Q_o R(z) - \int\limits_0^z \frac{dz'}{\lambda(z')} \int\limits_0^{z'} H(z'')dz'' + T_t(z) \tag{1}$$

where $T_o$ is the reference surface temperature, $Q_o$ the reference surface heat flux, the integral accounts for the vertical distribution of heat producing elements $H(z)$, and $T_t(z)$ is the temperature perturbation at depth $z$ due to time-varying changes to





the surface boundary condition. The thermal depth $R(z)$ is defined as:

$$R(z) = \int_0^z \frac{dz'}{\lambda(z')} \qquad (2)$$

where $\lambda$ is the thermal conductivity.

The temperature perturbation can be calculated by the following equation (Carslaw and Jaeger, 1959):

$$T_t(z) = \int_0^\infty \frac{z}{2\sqrt{\pi \kappa t^3}} \exp\left(\frac{-z^2}{4\kappa t}\right) T_o(t) dt \qquad (3)$$

where $\kappa$ is thermal diffusivity and $T_o(t)$ is the surface temperature at time $t$ before present. For a step change in surface temperature, $\Delta T$, at time t before present, the temperature perturbation $T_t(z)$ is given by Carslaw and Jaeger (1959):

$$T_t(z) = \Delta T \operatorname{erfc}\left(\frac{z}{2\sqrt{\kappa t}}\right) \qquad (4)$$

where erfc is the complementary error function. If the GST perturbations are approximated by their mean values $\Delta T_k$ during K intervals $(t_{k-1}, t_k)$, the temperature perturbation is written as follows:

$$T_t(z) = \sum_{k=1}^K \Delta T_k \left(\operatorname{erfc}\frac{z}{2\sqrt{\kappa t_k}} - \operatorname{erfc}\frac{z}{2\sqrt{\kappa t_{k-1}}}\right) \qquad (5)$$

$\Delta T_k$ is the average difference between the ground surface temperature during the time interval $(t_{k-1}, t_k)$ and the reference
surface temperature $T_0$.

### 2.1   Inversion

To reconstruct the GST history for each temperature-depth profile, we must invert equation 5. The inversion involves solving for the parameters $T_o$, $Q_o$, and $\Delta T_k$ of the temperature-depth profile. Equation 5 yields a system of linear equations in the unknown parameters for each depth where temperature has been measured. If $N$ temperature measurements were made in the
borehole, a system of $N$ linear equations with $K + 2$ unknowns, $T_o$, $Q_o$, and the $K$ values of $\Delta T_k$ is obtained. However this system of equations is ill-conditioned and its solution is unstable to small perturbations in the temperature data, i.e., a small error in the data results in a very large error in the solution (Lanczos, 1961). Different inversion methods are available to stabilize (regularize) the solution of ill posed problems (Backus-Gilbert method, Tikhonov regularization algorithm, Bayesian methods, singular value decomposition, Monte-Carlo methods). All these inversion techniques have been applied to reconstruct the GST
history (e.g., Vasseur et al., 1983; Nielsen and Beck, 1989; Shen and Beck, 1991; Mareschal and Beltrami, 1992; Clauser and Mareschal, 1995; Mareschal et al., 1999). In this paper, we have used the singular value decomposition because it is a very simple method to reduce the impact of noise and errors on the solution (Lanczos, 1961). This technique is well documented



for geophysical studies (Jackson, 1972; Menke, 1989) and its application for inversion of the ground temperature history is straightforward (Mareschal and Beltrami, 1992).

For sites including several boreholes with similar surface conditions, the data are inverted simultaneously because it is assumed that they have experienced the same surface temperature variations and therefore consistent subsurface temperature anomalies. It was expected that consistent trends in the temperature profiles would reinforce each other while errors and random noise would cancel each other. However, the resulting improvement in the signal to noise ratio remains marginal unless a sufficiently large number of profiles with the same GST history are available, which is almost never the case. Simultaneous inversion is described in detail and discussed by Beltrami and Mareschal (1992), Clauser and Mareschal (1995), and Beltrami et al. (1997), among others.

## 3 Description of data

Figure 1 shows the locations of the thirteen sites including twenty-five boreholes across northern Ontario and Quebec. The heat flow of these sites has previously been studied and a detailed description of the measurement techniques and sites can be found in the heat flow publications (Jessop, 1968; Jessop and Lewis, 1978; Lévy et al., 2010; Jaupart et al., 2014). All the sites are located north of 51°N, west and east of James Bay, and the boreholes range in depth between 400 and 800 m. All the holes, except Otoskwin and Nielsen Island, were drilled for mining exploration purposes. The temperature was measured at 10 m intervals using a calibrated thermistor. The sampling rate is higher at Otoskwin with temperature measurements every 1 m, while Nielsen Island was measured every 30 m. The overall accuracy is estimated on the order of 0.02 K with a precision of greater than 0.005 K. Thermal conductivity was measured on core samples by the method of divided bars (Misener and Beck, 1960). Radiogenic heat production measurements were also made on core samples but are not needed for corrections because the holes are not deep and the heat production rate is low.

Only eighteen holes proved suitable for inversion of the ground surface temperature. Their location and depth can be found in Table 1. Four sites are located in northern Ontario in a region of extensive discontinuous permafrost: Musselwhite, Thierry Mine, Otoskwin, and Noront. Thierry Mine (0605, 0606, 0608) and Noront (1012, 1013, 1014, 1015) include several boreholes. Six sites are located in northern Quebec in a region of sporadic discontinuous permafrost (Nielsen Island, LaGrande, Eastmain, Eleonore, Corvet, Camp Coulon), with Eastmain (0803, 0804) and Camp Coulon (0712, 0713, 0714) having multiple boreholes. Systematic variations in thermal conductivity observed at Nielsen Island were corrected by using the thermal depth (Bullard, 1939). Some measurements were not used for this study for different reasons (Table 2). The profile at Miminiska Lake (ONT) is too shallow to be inverted; the boreholes at Clearwater (QC) are plunging under a lake that affects the temperature profiles; the borehole at Poste Lemoyne is on the side of a very steep hill and the profile is seriously perturbed by the topography. We also discarded one of the temperature profiles at the Eleonore site because the borehole was plunging under a recently filled water reservoir and one of the profiles at the LaGrande site because it was a few m away from the edge of a 30 m cliff. The



borehole temperature-depth profiles at sites with multiple boreholes were truncated at the depth of the shallowest borehole to ensure that the same period of time was being studied (Thierry Mine at 530 m, Noront at 400 m, Eastmain at 400 m, and Camp Coulon at 400 m) (Beltrami et al., 2011). The temperature anomaly for each site was calculated by subtracting from the data the estimated steady-state temperature obtained by least-square fitting of a linear function to the bottom 100 m of the profile

(Figures 2-3).

## 4 Results

The temperature-depth profiles from the ten sites were inverted to reconstruct the GST histories for the last 500 years divided in intervals of 20 years (Figures 4-6). Simultaneous inversion was used at sites with multiple boreholes, Thierry Mine, Noront, Eastmain, and Camp Coulon. The cutoff value or number of eigenvalues determines which part of the solution is eliminated to

reduce the impact of noise. A lower cutoff value results in higher resolution in the reconstruction of the GST but at the expense of stability (Mareschal and Beltrami, 1992). Three eigenvalues (0.2 cutoff) were retained for all the sites except Otoskwin and Corvet, where four eigenvalues (0.08 cutoff) were retained. The results of the inversions are summarized in Table 3. Although the ground surface temperature histories differ in their details, they consistently show a trend of warming relative to the reference temperature (i.e. temperature 500 yrs before logging). Only one site shows indications that the GST was affected by the

LIA, a cold period that occurred between 200-500 yrs BP.

In northern Ontario, the trends of the inferred GST differ between sites. Otoskwin is the only site to show a LIA signal, with a cooling of ∼0.5 K with respect to the reference temperature (500 yrs BP). Evidence for this cooling can be found in the temperature anomaly at ∼200 m (Figure 2). Moreover, there is a noticeable change in the Otoskwin temperature gradient

at ∼200 m, which cannot be correlated to variations in thermal conductivity measured on 80 samples from the borehole. The recent warming at Musselwhite and Otoskwin occurred around the same time but it is observed earlier (∼250-300 yrs BP) at Thierry Mine and Noront (Figure 4). The total amplitude of warming differs greatly between the sites: 0.50 K with respect to reference temperature at Otoskwin, 0.88 K at Musselwhite, 1.85 K at Noront and 2.85 K at Thierry Mine. It is likely that the Thierry Mine signal was amplified by the clearing of vegetation that took place during the operation of the mine between 1934

and 1950.

Unlike for northern Ontario, a LIA signal was not found for any of the northern Quebec sites (Figures 5-6). A LIA signal was expected because pollen data have suggested that the cooling during the LIA (up to -0.3°C for North America) was strongest in northern Quebec (Gajewski, 1988; Viau and Gajewski, 2009; Viau, 2012). The onset of the recent climate warm-

ing is the same for all the sites (∼100-150 yrs BP), except Eleonore, where it began ∼200-300 yrs BP. The amplitude of the warming varies between 0.5-2 K (Figures 5-6) with the largest warming occuring at Corvet (2.18 K).





## 5  Discussion and Conclusions

Borehole temperature profiles in northern Ontario and Quebec consistently show a ground surface temperature increase of ∼1-2 K above the reference temperature. Most of this increase took place during the last 150 yrs. Two sites, Thierry Mine and Corvet appear to have large than average warming signals, 2.85 K and 2.18 K, respectively. The area around the Thierry Mine boreholes (0605, 0606, 0608) was cleared in the 1940s after the first opening of the mine. The change in vegetation cover could explain the enhanced warming signal (Lewis and Wang, 1998; Lewis, 1998). Corvet is located on the side of a 30 m hill. It has

been shown that topography distorts the temperature isotherms (Jeffreys, 1938): a positive topography leads to a reduced temperature gradient and an increased apparent warming signal (e.g., Blackwell et al., 1980; Guillou-Frottier et al., 1998). These examples further illustrate the significant influence of non-climatic effects on ground surface temperature reconstructions from borehole temperature-depth profiles.

A cooling period corresponding to the LIA was found for only one site, Otoskwin (ON), which exhibits marked perturbations of the temperature profile. While spatial and temporal variation in the LIA have been noted (Matthews and Briffa, 2005), the absence of a consistent LIA signal in northern Ontario and Quebec deserves some discussion. The LIA cooling period has been inferred from different proxies and selected borehole temperature-depth profiles in eastern Canada (e.g., Archambault and Bergeron, 1992; Beltrami and Mareschal, 1992; Wang and Lewis, 1992; Chouinard et al., 2007; Bunbury et al., 2012). For

example, pollen data indicate a pronounced LIA cooling in Quebec (Viau and Gajewski, 2009; Gajewski, 1988). The lack of LIA signal in the majority of the borehole inversions could be related to a combination of several factors. One is the limited resolution of the inversion of borehole temperature profiles. In the presence of noise, a period of weak cooling between 500 and 200 yrs B.P. followed by strong warming is difficult to resolve. Resolution at Otoskwin is better because the singular value cutoff was lowest at this site. Also, Chouinard and Mareschal (2007) suggested that the LIA could have started ∼100 yrs earlier

in northern Quebec than in southern Canada. Resolving the LIA would require a borehole deeper than ∼400 m, which is not the case of all the holes. Let us also point out that the sampling resolution at Nielsen Island is low, with measurements only every 30 m, which is not sufficient to resolve a LIA signal. While the absence of LIA signal in Quebec was unexpected, its absence in northern Ontario confirms the findings of Gosselin and Mareschal (2003), who found only 2 sites with a LIA signal among 33 temperature-depth profiles from northwestern Ontario. They hypothesized that the lack of LIA signal could be due to the

influence of Lake Superior because the two sites with LIA signals were above 50°N and the furthest from Lake Superior. This is not supported by the present study because the 4 Ontario sites are several hundreds of kilometres away from Lake Superior. It is also possible that the LIA signal is masked by other physical effects, such as an advance and retreat of permafrost or a change in the precipitation regime and the duration of the ground snow cover during the LIA.

No geographic trends in the GST histories were observed, despite different SAT conditions. Meteorological data from the NOAA weekly dataset and 8 general circulation models (GCMs) for the period of 1970-1999 display a longer snow cover duration in northern Quebec than in northern Ontario (Brown and Mote, 2009). The higher precipitation is confirmed by proxy



reconstructions of lake levels and tree forms (Bégin, 2000; Lavoie and Payette, 1992). Because of the greater snowfall and longer snow cover, the present ground surface relative to air surface temperatures in Quebec are warmer than in Ontario. However these dissimilar conditions have not resulted in noticeable discordance between the GST histories between northern Ontario and Quebec, suggesting that the same differences in precipitation persisted throughout the period reconstructed.

The magnitude of the recent warming is about the same as the ∼1-2 K warming for the past 150 years inferred from several studies in the southern portion of the Superior Province (Beltrami and Mareschal, 1992; Shen and Beck, 1992; Chouinard and Mareschal, 2007) and less than the very pronounced warming in the eastern Canadian Arctic (Beltrami and Taylor, 1995; Taylor et al., 2006; Chouinard et al., 2007).

The sites are located in a region described as discontinuous permafrost, where ground temperatures are slightly below freezing, at least according to the Canadian and world permafrost maps. No sign of permafrost was found at any of the measured sites nor at the sites that were excluded (see Table 2). Not only are the present average ground surface temperatures well above the freezing point of water, but also, except for Nielsen Island, the ground surface temperature histories retrieved from inversion reveal that the temperature has remained well above the freezing point for the last 500 years. Furthermore, during logging of more than 100 holes in the regions of Manitoba, Saskatchewan, and northern Ontario classified as extensive discontinuous permafrost, permafrost has been encountered at only one hole, north of the town of Lynn Lake in northern Manitoba (Guillou-Frottier et al., 1998). Clearly, the spatial distribution of permafrost outlined in the available permafrost maps is inaccurate and thus unreliable possibly because they are not based on sufficiently deep ground temperature measurements but estimated from interpolated sparse records of SAT (Heginbottom, 2002; Gruber, 2012). The discrepancy between permafrost maps and direct field observations reveal that SAT interpolations are unsuitable to estimate the spatial variations of ground temperatures. This is likely because the maximum thickness of snow exceeds 1 m at the end of the winter remaining on the ground from mid December to mid April, resulting in a large offset between the GST and SAT (Zhang, 2005; Grosse et al., 2016). Our study suggests that borehole temperature profiles could be used in the future to assess the reality of the permafrost retreat assumed to have occurred after the LIA (Halsey et al., 1995; Schuur et al., 2008). Furthermore, borehole temperature profiles might be a better means for determining the southern extent of areas of past and present permafrost than current permafrost maps and a useful tool for validation of climate models.

*Acknowledgements.* This work was supported by grants from the Natural Sciences and Engineering Research Council of Canada Discovery Grant (NSERC DG 140576948) and the Canada Research Program (CRC 230687) to H. Beltrami. Computational facilities provided by the Atlantic Computational Excellence Network (ACENnet-Compute Canada) with support from the Canadian Foundation for Innovation. H. Beltrami holds a Canada Research Chair in Climate Dynamics. C.Pickler is funded by graduate fellowships from a NSERC CREATE Training Program in Climate Sciences based at St.Francis Xavier University.



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







**Figure 1.** Map of Ontario and western Quebec showing the location of sites (red dots). For sites with several boreholes (Camp Coulon , Eastmain, Thierry Mine, and Noront), the number of profiles available is enclosed in parenthesis. Black diamonds show the locations of sites that were discarded.




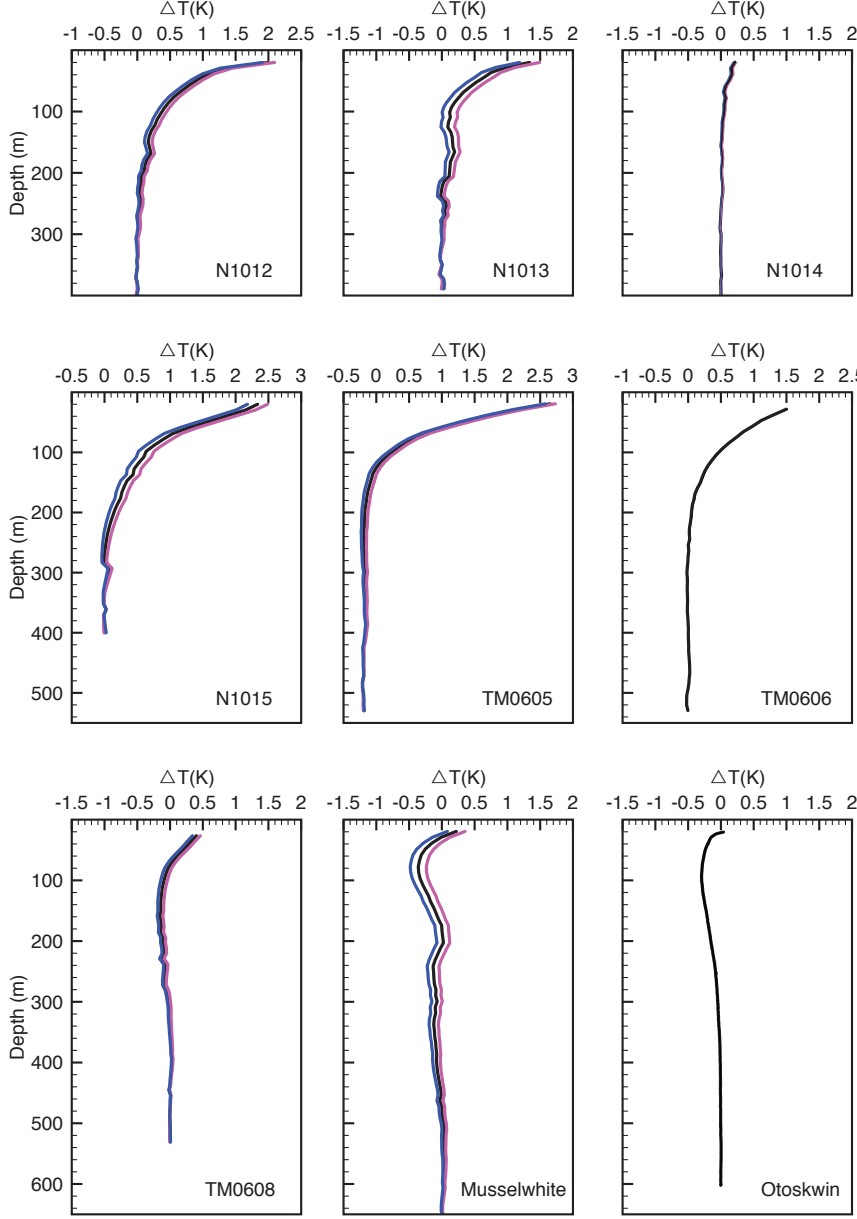

**Figure 2.** Temperature anomalies for the northern Ontario boreholes. Holes TM0605, TM0606, and TM0608 are from the Thierry Mine site; holes N1012, N1013, N1014, and N1015 belong to the Noront site. The anomaly is obtained by subtracting the estimated steady-state geotherm obtained by the least-square fit of a straight line to the bottom 100 m of the borehole temperature-depth profile. The black line represents the best linear fit, while the pink and blue lines are the upper and lower bounds, respectively, of the $2\sigma$ confidence intervals. For N1014, TM0606, and Otoskwin, the upper and lower bounds of the confidence interval are not visible due to the temperature scale. The temperature anomaly at Musselwhite was cut at 650 m.




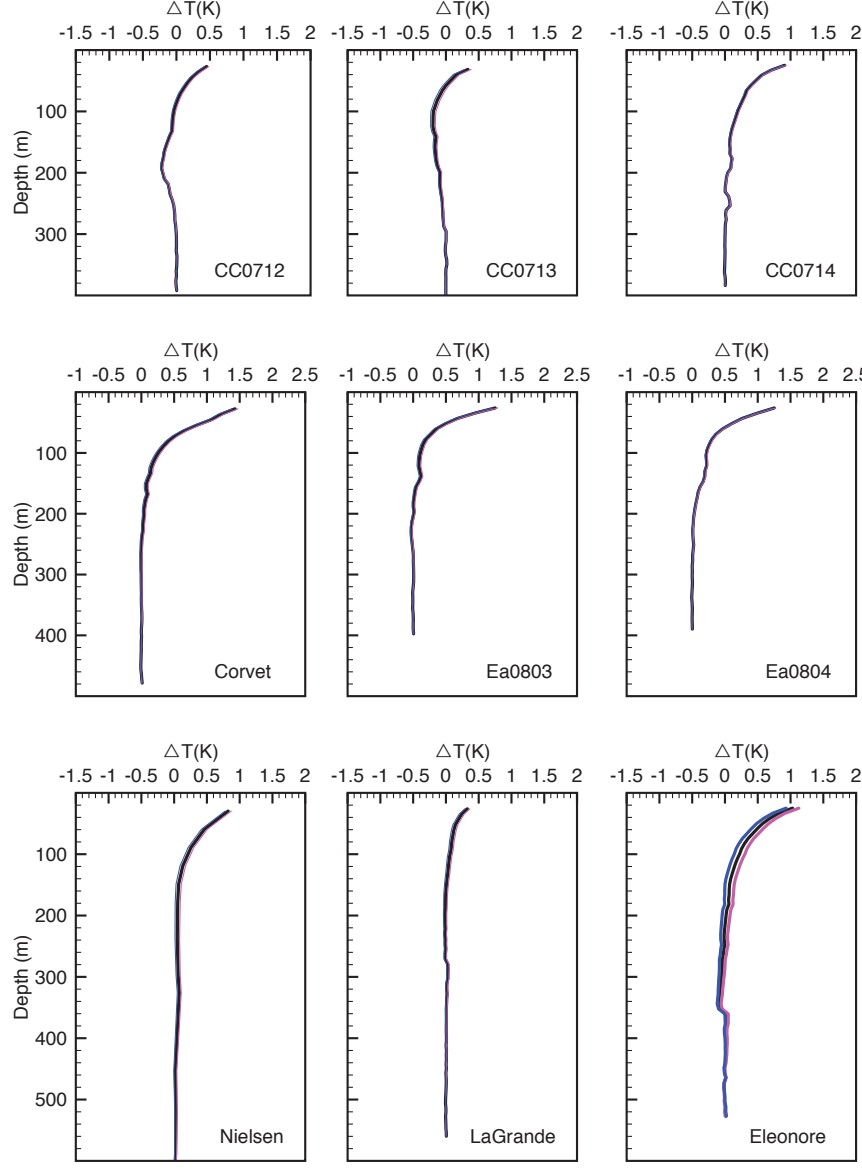

**Figure 3.** Temperature anomalies for the northern Quebec boreholes. CC0712, CC0713, CC0714 are the boreholes from Camp Coulon; Ea0803 and Ea0804 are the boreholes from Eastmain . The anomaly is obtained by subtracting the estimated steady-state geotherm obtained by the least-square fit of a straight line to the bottom 100 m of the borehole temperature-depth profile. The black line represents the best linear fit, while the pink and blue lines represent the upper and lower bounds, respectively, of the $2\sigma$ confidence intervals. For N1014, TM0606, and Otoskwin, the upper and lower bounds of the confidence interval are not visible due to the temperature scale. The temperature anomaly at Nielsen Island was cut at 600m.



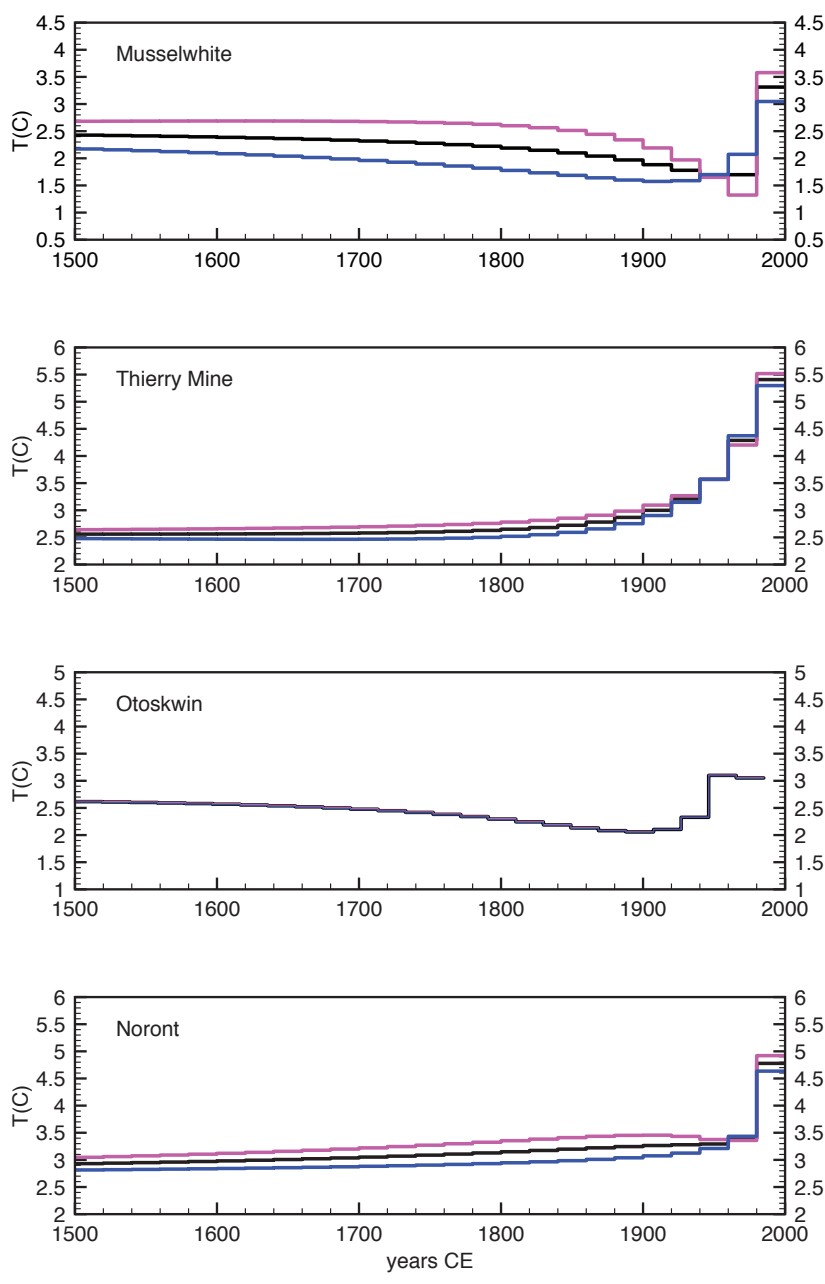

**Figure 4.** GST histories for the northern Ontario sites determined by inversion of the anomalies. For multiple holes at a given site (Thierry mine and Noront), simultaneous inversion was used. The pink and blue lines represent the inversions of the upper and lower bounds of the anomaly. For Otoskwin, the three lines are superposed.



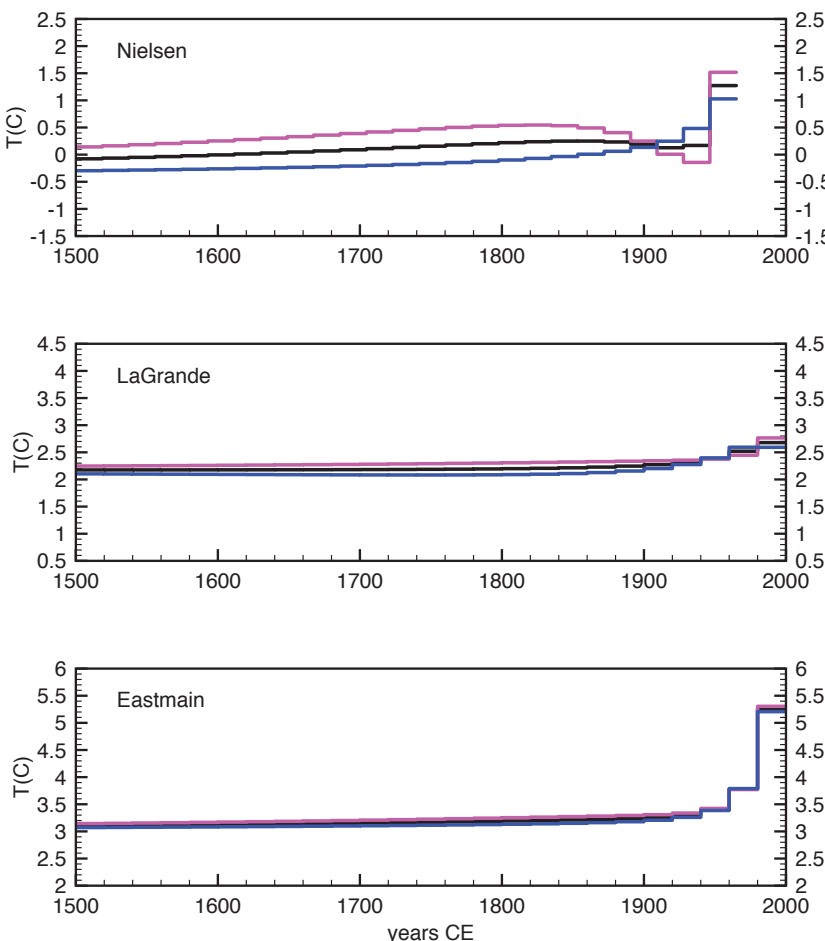

**Figure 5.** GST histories for the northern Quebec sites. Simultaneous inversion was used for Eastmain, which includes two holes. The pink and blue lines represent the inversions of the upper and lower bounds of the anomaly.



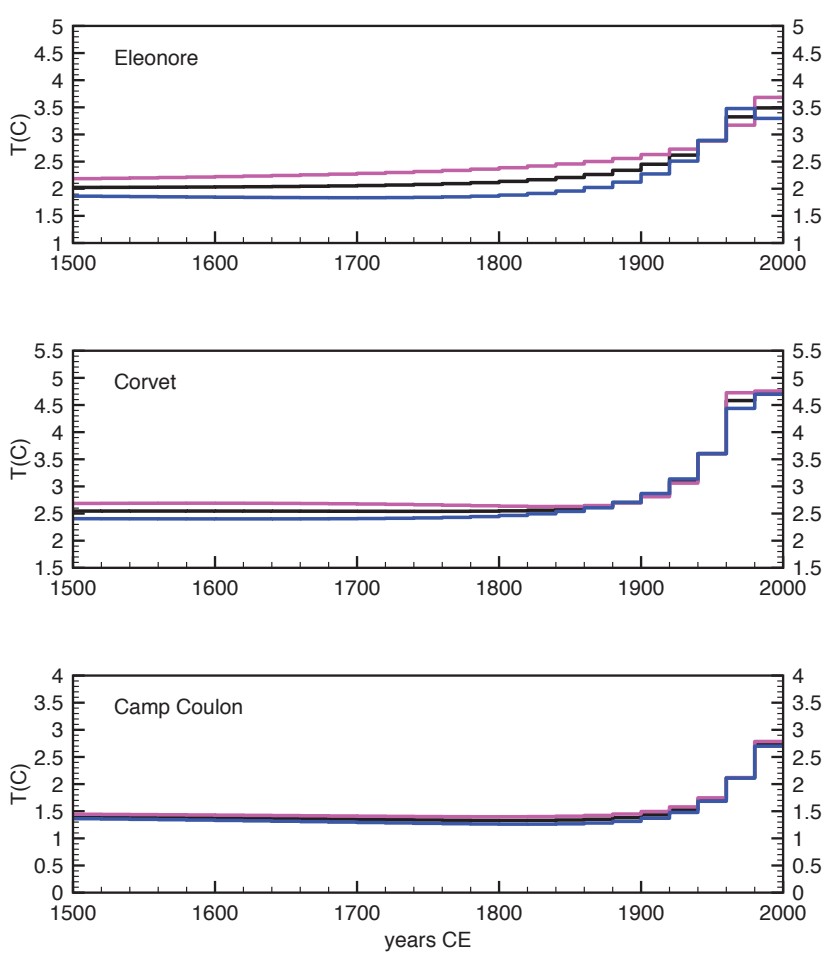

**Figure 6.** GST histories for the northern Quebec sites. Simultaneous inversion was used for Camp Coulon, which includes more than one

hole. The pink and blue lines represent the inversions of the upper and lower bounds of the anomaly.





**Table 1.** Location and technical information concerning the boreholes used in this study, where $\lambda$ is the thermal conductivity, Q is the heat flux, and $Q_{corr}$ is the heat flux corrected for post glacial warming

| Site | Log ID | Latitude | Longitude | Depth (m) | $\lambda$ (W m$^{-1}$ K$^{-1}$) | Q (mW m$^{-2}$) | $Q_{corr}$ (mW m$^{-2}$) | Reference |
|------|--------|----------|-----------|-----------|------------------|-----------------|---------------------------|-----------|
| Musselwhite | 0601 | 52°37′28.3″ | 90°23′32.7″ | 740 | 2.73 | 30.7 | 33 | Lévy et al. (2010) |
| Thierry Mine | | | | | | | **28** | |
| | 0605 | 51°30′24″ | 90°21′11″ | 802 | 2.91 | 26.8 | 29.7 | Lévy et al. (2010) |
| | 0606 | 51°30′22″ | 90°21′11″ | 530 | 2.61 | 24.4 | 27.0 | Lévy et al. (2010) |
| | 0608 | 51°30′24.3″ | 90°21′10.7″ | 737 | 2.60 | 24.6 | 27.1 | Lévy et al. (2010) |
| Otoskwin | | | | | | | **25** | |
| | n/a | 51°49.5′ | 89°35.9′ | 602 | 2.79 | 20 | 25 | Jessop and Lewis (1978) |
| Noront | | | | | | | **35** | |
| | 1012 | 52°44′23.9″ | 86°17′42.1″ | 761 | 3.2 | 34.1 | 37.6 | Jaupart et al. (2014) |
| | 1013 | 52°44′25″ | 86°17′42.1″ | 389 | 3.1 | 28.3 | 34.1 | Jaupart et al. (2014) |
| | 1014 | 52°44′29.9″ | 86°17′59.9″ | 762 | 3.1 | 31.7 | 35.6 | Jaupart et al. (2014) |
| | 1015 | 52°44′25″ | 86°17′42.1″ | 806 | 3.1 | 29.5 | 33.4 | Jaupart et al. (2014) |
| Nielsen Island | | | | | | | **26** | |
| | 0-77 | 55°23.7′ | 77°41.0′ | 1408 | 5.5 ($\leq$ 400m) 2.4 ($>$ 400 m) | - | 26.8 | Jessop (1968) |
| LaGrande | | | | | | | **22** | |
| | 0405 | 53°31′45″ | 76°33′15″ | 600 | 2.89 | 19.2 | 21.9 | Lévy et al. (2010) |
| Eastmain | | | | | | | **34** | |
| | 0803 | 52°10′16″ | 76°27′66″ | 398 | 2.9 | 28.8 | 34.1 | Jaupart et al. (2014) |
| | 0804 | 52°10′16″ | 76°27′66″ | 390 | 2.9 | 26.7 | 32.9 | Jaupart et al. (2014) |
| Eleonore | | | | | | | **31** | |
| | 0502 | 52°42′05″ | 76°04′46″ | 527 | 2.47 | 30.7 | 32.8 | Lévy et al. (2010) |
| Corvet | | | | | | | **27** | |
| | 0716 | 53°19′72″ | 73°55′60″ | 479 | 2.80 | 24.0 | 26.8 | Lévy et al. (2010) |
| Camp Coulon | | | | | | | **28** | |
| | 0712 | 54°47′43″ | 71°17′09″ | 561 | 3.69 | 26.4 | 29.0 | Lévy et al. (2010) |
| | 0713 | 54°47′95″ | 71°17′20″ | 460 | 3.73 | 24.5 | 27.5 | Lévy et al. (2010) |
| | 0714 | 54°47′43″ | 71°17′34″ | 384 | - | - | - | Lévy et al. (2010) |





**Table 2.** Location and technical information concering boreholes not suitable for this study, where $T_o$ is the reference surface temperature and $Q_o$ is the reference heat flux

| Site | Log ID | Latitude | Longitude | $T_o$ | $Q_o$ | Remark |
|------|--------|----------|-----------|-------|-------|--------|
| | | | | (°C) | (mW m$^{-2}$) | |
| Miminiska | 0603 | 51°34′51″ | 88°31′09″ | 3.33±0.02 | 25.8±1.5 | Too shallow |
| LaGrande | 0406 | 53°31′42″ | 76°33′49″ | 2.67±0.005 | 14.1±0.3 | Topography |
| Eleonore | 0503 | 52°42′00″ | 76°04′45″ | 2.56±0.01 | 27.7±0.6 | Reservoir |
| Clearwater | 0505 | 52°12′33″ | 75°48′38″ | 2.33±0.02 | 30.6±0.4 | Lake |
| | 0506 | 52°12′31″ | 75°48′23″ | 2.23±0.01 | 31.3±0.2 | Lake |
| | 0507 | 52°12′39″ | 75°48′23″ | 2.73±0.004 | 30.5±0.3 | Lake |
| Poste Lemoyne | 0715 | 53°27′37″ | 75°12′21″ | 1.83±0.004 | 24.5±0.2 | Topography |




**Table 3.** Summary GST History Results where $T_o$ is the reference surface temperature, $Q_o$ is the reference heat flux, $\Delta T$ is the difference between the maximal temperature and the reference temperature 500 yrs before logging.

| Site | Log ID | year | $T_o$ (°C) | $Q_o$ (mW m$^{-2}$) | $\Delta T$ (K) | LIA |
|---|---|---|---|---|---|---|
| Musselwhite | 0601 | 2006 | 2.56±0.01 | 30.0±0.5 | 0.88 | no |
| Thierry Mine | | | | | 2.85 | no |
| | 0605 | 2006 | 2.63±0.01 | 25.9±0.5 | | |
| | 0606 | 2006 | 2.55±0.01 | 23.5±0.7 | | |
| | 0608 | 2006 | 2.62±0.01 | 23.9±0.3 | | |
| Otoskwin | n/a | 1985 | 2.81±0.001 | 19.5±0.05 | 0.50 | yes |
| Noront | | | | | 1.85 | no |
| | 1012 | 2010 | 1.16±0.01 | 35.5±0.9 | | |
| | 1013 | 2010 | 1.51±0.02 | 31.3±1.5 | | |
| | 1014 | 2010 | 1.78±0.004 | 32.2±0.3 | | |
| | 1015 | 2010 | 2.35±0.02 | 27.6±1.4 | | |
| Nielsen Island | 0-77 | 1977 | -0.43±0.02 | 23.7±0.4 | 1.35 | no |
| LaGrande | 0405 | 2004 | 2.18±0.004 | 19.4±0.2 | 0.50 | no |
| Eastmain | | | | | 2.15 | no |
| | 0803 | 2008 | 2.98±0.01 | 27.3±0.4 | | |
| | 0804 | 2008 | 3.01±0.003 | 27.0±0.2 | | |
| Eleonore | 0502 | 2005 | 2.04±0.01 | 30.9±0.5 | 1.46 | no |
| Corvet | 0716 | 2007 | 2.45±0.01 | 24.9±0.3 | 2.18 | no |
| Camp Coulon | | | | | 1.34 | no |
| | 0712 | 2007 | 1.46±0.01 | 26.6±0.4 | | |
| | 0713 | 2007 | 1.83±0.01 | 24.2±0.6 | | |
| | 0714 | 2007 | 1.33±0.004 | - | | |