# Peer review of "Climate trends in northern Ontario and Quebec from borehole temperature profiles"

_Climate of the Past, 2016_

## Short Comment (SC1) · 29 Jun 2016

R. Way

rway024@uottawa.ca

This is a brief comment on 'Climate trends in northern Ontario and Quebec from borehole temperature profiles' by Pickler et al.

The authors incorrectly state that their sample sites are all located within the sporadic (10-50% of the land surface) to extensive discontinuous zones (50-90% of the land surface). In nearly every case the sampled sites are located within the isolated patches permafrost zone (<10% of land surface) according to maps produced by Heginbottom et al (1995), Payette (2001) and recent spatial numerical modelling of permafrost distribution for Labrador-Ungava (Way and Lewkowicz, 2016). Considering the more realistic permafrost extent, there is no discrepancy between the borehole observations and existing permafrost maps. A further point on a similar subject is that the temperature sampling methodology here is at too coarse a resolution (depth) to detect thin permafrost bodies if they were to exist. In the southern end of the discontinuous zone you would be more likely to find thinner permafrost bodies therefore this is a serious limitation of the study.

The article also does not present any indication of the land cover types encountered in the study area and correspondingly, does not consider how permafrost is distributed across the landscape (e.g. Shur and Jorgenson, 2007; Jorgenson et al., 2010). In northern Ontario and Québec, permafrost is largely absent from forested areas at the southern end of the discontinuous zones where snow accumulates while concurrently being present on wind exposed mountaintops (Brown, 1979; Ives, 1979; Allard and Séguin, 1987; Granberg, 1989; Ou et al. 2016a,b; Way and Lewkowicz, 2016). Ignoring these critical variables makes it untenable to draw large-scale conclusions on permafrost from the provided data.

In general, I believe that the discussion of permafrost in this article should be removed in its entirety as the methodology, discussion and interpretations presented are not appropriate for the analysis of permafrost distribution and history. Finally, the lack of consideration of the literature on permafrost in western Québec and northern Ontario must be addressed.

References Allard, M. and Séguin, M. K.: Le pergélisol au Québec nordique : bilan et perspectives, Géographie physique et Quaternaire, 41(1), 141, doi:10.7202/032671ar, 1987.

Brown, R. J.: Permafrost distribution in the southern part of the discontinuous zone in Quebec and Labrador, Géographie physique et Quaternaire, 33(3–4), 279–289, 1979.

Granberg, H. B.: Permafrost mapping at Schefferville, Québec, Physical Geography, 10(3), 249–269, 1989.

Heginbottom, J. A., Dubreuil, M.-A. and Harker, P. A.: Canada – Permafrost, [online]

Available from: http://geogratis.gc.ca/api/en/nrcan-rncan/ess-sst/d1e2048b-ccff-5852-aaa5-b861bd55c367.html, 1995.

Ives, J. D.: A proposed history of permafrost development in Labrador-Ungava, Géographie physique et Quaternaire, 33(3–4), 233–244, doi:10.7202/1000360ar, 1979.

Ou, C., LaRocque, A., Leblon, B., Zhang, Y., Webster, K. and McLaughlin, J.: Modelling and mapping permafrost at high spatial resolution using Landsat and Radarsat-2 images in Northern Ontario, Canada: Part 2 – regional mapping, International Journal of Remote Sensing, 1–29, doi:10.1080/01431161.2016.1151574, 2016.

Ou, C., Leblon, B., Zhang, Y., LaRocque, A., Webster, K. and McLaughlin, J.: Modelling and mapping permafrost at high spatial resolution using Landsat and Radarsat images in northern Ontario, Canada: part 1 – model calibration, International Journal of Remote Sensing, 1–24, doi:10.1080/01431161.2016.1157642, 2016.

Payette, S.: Les processus et les formes périglaciaires, in Écologie des tourbières du Québec-Labrador, pp. 199–239, Les Presses de l'Université Laval, Québec City, Canada., 2001.

Shur, Y. L. and Jorgenson, M. T.: Patterns of permafrost formation and degradation in relation to climate and ecosystems, Permafrost and Periglacial Processes, 18(1), 7–19, doi:10.1002/ppp.582, 2007.

Jorgenson, M. T., Romanovsky, V., Harden, J., Shur, Y., O'Donnell, J., Schuur, E. A. G., Kanevskiy, M. and Marchenko, S.: Resilience and vulnerability of permafrost to climate change, Canadian Journal of Forest Research, 40(7), 1219–1236, doi:10.1139/X10-060, 2010.

Way, R. G. and Lewkowicz, A. G.: Investigations of discontinuous permafrost in coastal Labrador with DC electrical resistivity tomography, in Proceedings of GéoQuebec: 68th Canadian Geotechnical Conference and 7th Canadian Permafrost Conference, p. 8, Québec City, Canada., 2015.

Way, R. G. and Lewkowicz, A. G.: Modelling the spatial distribution of permafrost in Labrador-Ungava using TTOP, Canadian Journal of Earth Sciences, doi:10.1139/cjes-2016-0034, 2016.
* * *

---

## Editor Comment (EC1) · A. Winguth (Editor) · 30 Jun 2016

Dear Dr. Way,

Thank you very much for your comments on manuscript cp-2016-55.

Sincerely, Arne Winguth

Editor

---

## Author Comment (AC1) · 11 Jul 2016

We thank R.Way for his comment and would like to address several points that he brought up.

1. *The authors incorrectly state that their sample sites are all located within the sporadic (10-50% of the land surface) to extensive discontinuous zones (50-90% of the land surface). In nearly every case the sampled sites are located within the isolated patches permafrost zone (<10% of land surface) according to maps produced by Heginbottom et al (1995), Payette (2001) and recent spatial numerical modelling of permafrost distribution for Labrador-Ungava (Way and Lewkowicz, 2016). Considering the more realistic permafrost extent, there is no discrepancy between the borehole observations and existing permafrost maps.*

We thank Mr Way for pointing out that the permafrost map that we used is inaccurate. We agree that, based on the permafrost map of the Geological Survey of Canada, we must qualify our conclusions. In Quebec, we cover a region classified as having only isolated patches of permafrost. We did not find permafrost at any site but our statistics is not sufficient to draw any conclusion. In northern Ontario, one site (4 holes) is near the southern edge of a region with extensive discontinuous permafrost (50-90%). Using the low frequency of 50%, the probability that not one of 4 holes meets permafrost is low: 1/16 or 6.25%. The other sites in Ontario are indeed irrelevant to permafrost.

In Manitoba, we had sampled a region of sporadic discontinuous permafrost (10-50%) between the towns of Flin Flon, Thompson, and Lynn Lake, but we found that only 1 hole in more than 60 was blocked, as described and discussed by Guillou-Frottier et al. (1998). Using the lowest value of 10% permafrost coverage, the probability that only one hole out of 60 encounters permafrost is only $0.1 \times 0.9^{59} \times 60$ or 1.20%; for the 50% frequency, the probability is only $5.2 \times 10^{-17}$!

2. *A further point on a similar subject is that the temperature sampling methodology here is at too coarse a resolution (depth) to detect thin permafrost bodies if they were to exist. In the southern end of the discontinuous zone you would be more likely to find thinner permafrost bodies therefore this is a serious limitation of the study.*

We disagree. To avoid repetition of the measurement procedure, we did not sufficiently emphasize the obvious fact that the temperature measurements in boreholes must be performed a long time after drilling when the ground has returned to thermal equilibrium. Should permafrost be present, the hole would be frozen and we would not be able to lower the probe in it to measure temperature. There is no permafrost in the holes that we measured regardless of the depth sampling interval. A second point is that thin permafrost bodies are very unlikely to be present for simple reasons of thermodynamics. The second law of

thermodynamics implies that any temperature oscillation decays with time, and it follows that in steady state the vertical temperature profile can have no maximum or minimum, implying that the existence of a thin frozen layer is impossible in steady state. A frozen underground layer could exist but only as a transient. The life time of such a layer depends on its thickness. For standard values of thermal diffusivity, a 1m thick layer would last less than 1 year.

3. *The article also does not present any indication of the land cover types encountered in the study area and correspondingly, does not consider how permafrost is distributed across the landscape (e.g. Shur and Jorgenson, 2007; Jorgenson et al., 2010). In northern Ontario and Québec, permafrost is largely absent from forested areas at the southern end of the discontinuous zones where snow accumulates while concurrently being present on wind exposed mountaintops (Brown, 1979; Ives, 1979; Allard and Séguin, 1987; Granberg, 1989; Ou et al. 2016a,b; Way and Lewkowicz, 2016). Ignoring these critical variables makes it untenable to draw large-scale conclusions on permafrost from the provided data.*

It is correct that we did not consider the landscape in these studies. The main reason being that we use holes of opportunity that were drilled for mining exploration in any surface environment. They represent thus a random unbiased sampling of the landscape. Outside Quebec, the study area is in lowlands without "mountaintops". The topography near our sites on the Quebec side is more marked but remains low. Most likely, disagreements arise from the inconsistent and poor quality of the permafrost data because of insufficient sampling and the extreme spatial variability of the land surface.

4. *In general, I believe that the discussion of permafrost in this article should be removed in its entirety as the methodology, discussion and interpretations presented are not appropriate for the analysis of permafrost distribution and history. Finally, the lack of consideration of the literature on permafrost in western Québec and northern Ontario must be addressed.*

We appreciate the comment but must disagree. The methodology is sound, and our reconstruction of past temperature histories is based on very simple physics. As much as we understand the need for considering the literature on permafrost, we believe that permafrost studies would gain to be confronted with physical models. A dialogue would be more beneficial than outright dismissal!

**References**

Guillou-Frottier, L., Mareschal, J. C., and Musset, J.: Ground surface temperature history in central Canada inferred from 10 selected borehole temperature profiles, Journal of Geophysical Research, 103, 7385–7397, 10.1029/98JB00021, 1998.

---

## Referee Comment (RC1) · Anonymous Referee #1 · 16 Sep 2016

This manuscript discusses several borehole temperature profiles in portions of two Canadian provinces. The authors use an inversion technique commonly used in borehole climate studies to determine past temperature histories for these sites, and interpret the results as showing little or no signal for both the Little Ice Age (LIA) and the presence of permafrost in the region.

My overall feeling of the paper is positive, but I believe that the authors gloss over several issues that either need to be addressed or that would make the manuscript, and hence the interpretations, stronger. Addressing these issues would lead me to recommend publication of this manuscript in Climate of the Past.

I will list my comments about these issues below:

1. Borehole sites: In adequately determining if borehole sites are appropriate for use

in climate reconstructions, several criteria are required. While the authors have addressed several of the sites and determined they were unsuitable (as presented in Table 2 of the manuscript), information regarding the other sites is not included that would aid a reader in understanding the conditions at the boreholes. For example, no discussion of slope, topography, vegetation or surface material is given, although the authors do reference previous studies. A discussion of vegetation and ground cover at the sites would be extremely useful, however, especially considering that the argument that one site (Thierry Mine) may have additional warming due to the removal of vegetation was put forth. Further, some sites are said to be "too shallow" or on the side of "steep" hills. What exactly is "too shallow" and "steep"? Can a quantitative discussion replace the qualitative explanation? Also, are all of the boreholes vertical? At least one site was excluded because it was plunging under a lake. It should be clear.

2. Temperature Anomalies: The determination of the geothermal gradient using the bottom 100 meters is usually sufficient, but I wonder if the results of the removal of the steady state gradient as shown in Figures 2 and 3 are different if the length (100 m) is modified?

3. Results: The authors state that only one site has a ground surface temperature (GST) that was affected by the LIA. However, based on the temperature anomalies shown in Figures 2 and 3, it would seem that other sites exhibit cooling at the same depths as Otoskwin. Mussellwhite, TM0608, and CC0713 all have temperature anomalies that indicate cooling at the same general depth. Is this not a LIA signature? Also, the anomaly shown in CC0712 (Figure 3, top left) has a very interesting profile. What is the cause? Other questions I have about the results that don't have any explanation – or that aren't adequately explained – include the assertion that the Thierry Mine signal may be amplified by the clearing of vegetation between 1934 and 1950. However, most of the GST histories show a large increase in temperature at this same time, indicating it may not be vegetation alone. Have the authors done any modeling or do they have any surface temperatures to support this hypothesis? Lastly, one site (Eleonore) has

warming that began considerably earlier than the other sites. Why might this be?

4. Interpretations: In the final section, the authors attempt to explain some of the anomalous results, but do so only qualitatively. The manuscript would be far stronger if there was more of an attempt to quantitatively make the same argument. This is done with the Thierry Mine example (previously discussed), as well as with Corvet, which "is located on the side of a 30 m hill." However, what is the slope? How much of an effect does this have? It is still being used, so the authors must think it isn't significant. My final criticism is of the qualitative nature of the LIA arguments. The authors discuss what the LIA surface signal should be for the region, but do not see a ground signal. Perhaps a simple forward model of driving into the ground a surface temperature time series with the appropriate LIA signal and making a comparison to the boreholes would be appropriate? Then, the authors could argue whether the signal is strong enough to actually be observed, or whether it is not seen due to snow or something else. This is similar to the arm waving argument used to interpret a possible ground warming due to longer/deeper snow cover in the region, but it seems that other authors have performed some analysis that may provide quantitative support to their arguments (perhaps Bartlett et al., 2005?)

5. References: I did notice that on page 11 in the reference section that Jaupart and Mareschal, 2011 was published in Cambridge, not Cabridge; also, the next two references following the previous reference are of Jaupart et al., 2014 and are a duplicate.

---

## Author Comment (AC2) · 22 Sep 2016

We thank the reviewer for his thoughtful and constructive comments. We do agree with several points that he/she has made and we shall include his/her suggestions in the revised manuscript.

1. *Borehole sites: In adequately determining if borehole sites are appropriate for use in climate reconstructions, several criteria are required. While the authors have addressed several of the sites and determined they were unsuitable (as presented in Table 2 of the manuscript), information regarding the other sites is not included that would aid a reader in understanding the conditions at the boreholes. For example, no discussion of slope, topography, vegetation or*

[Figure]

*surface material is given, although the authors do reference previous studies. A discussion of vegetation and ground cover at the sites would be extremely useful, however, especially considering that the argument that one site (Thierry Mine) may have additional warming due to the removal of vegetation was put forth. Further, some sites are said to be "too shallow" or on the side of steep hills. What exactly is too shallow and steep? Can a quantitative discussion replace the qualitative explanation? Also, are all of the boreholes vertical? At least one site was excluded because it was plunging under a lake. It should be clear.*

We logged the majority of the holes and noted in our log books if terrain conditions appeared to be not suitable. Holes less than 300 m were rejected for being too shallow. Holes were rejected for being near a lake when the mean distance was less than the depth of the hole, or less than 300 m. Holes were deemed too steep and rejected if they had slope of 5% or more over distance comparable to depth. The dip of the holes varies between 55° and 90°. Only two holes, Nielsen Island and Otoskwin, were not inclined. A more detailed description of each site will be provided.

2. *... if the results of the removal of the steady state gradient as shown in Figures 2 and 3 are different if the length (100 m) is modified?*

Jaupart et al. (2014) and Lévy et al. (2010) have analyzed the heat flow of the majority of the boreholes presented in this manuscript. They varied the length they used to calculate the steady state. Some differences are noted including the heat flow for the Thierry Mine sites differing by 5%. This could be related to the differing lengths used to calculate the steady state. But, inversions were done using the complete profile and no significant differences were noted with those reconstructed from the residual (only the temperature anomaly).

3.  *The authors state that only one site has a ground surface temperature (GST) that was affected by the LIA. However, based on the temperature anomalies shown in Figures 2 and 3, it would seem that other sites exhibit cooling at the same depths as Otoskwin. Mussellwhite, TM0608, and CC0713 all have temperature anomalies that indicate cooling at the same general depth. Is this not a LIA signature*

    All the temperature anomalies of the three sites (Musselwhite, TM0608, CC0713) indicate cooling. However, the profiles and anomalies at Musselwhite and TM0608 are noisy, which could mask a clear LIA signal when inverting the sites. Mareschal and Beltrami (1992) showed that resolution decreases when noise and errors must be filtered. Larger singular values are required when dealing with noisy data since they reduce the impact of noise but retain the gross features of the solution. If the LIA signal is too weak in a noisy profile, it will not resolved. The CC0713 temperature anomaly is less noisy and shows a mild cooling signal of $\leq 0.2$ K. The site was inverted individually and no LIA signal was observed since it is too small to be resolved. Furthermore, the choice of singular value could impact the ability to reconstruct a LIA signal. A test was run with a 1 K cooling between 1600 and 1800 and varying the singular value cutoff (Figure 1). For noise-free synthetic data, a 1 K cooling signal cannot be resolved with less than 5 singular values.

4.  *Also, the anomaly shown in CC0712 (Figure 3, top left) has a very interesting profile. What is the cause?*

    There are discontinuities in the gradient between 100 and 300 m, which could be due to small water flows.

5. *Other questions I have about the results that don't have any explanation or that aren't adequately explained include the assertion that the Thierry Mine signal may be amplified by the clearing of vegetation between 1934 and 1950. However, most of the GST histories show a large increase in temperature at this same time, indicating it may not be vegetation alone. Have the authors done any modeling or do they have any surface temperatures to support this hypothesis*

The warming at Thierry Mine is greater than the other sites (at least 0.7 K greater than any other site). Figure 2 locates the three Thierry Mine boreholes (0605, 0606, 0608). From the satellite image, the three Thierry Mine sites are ~500 m away from a large clearing. This clearing is associated with the development of the nearby mine in 1934-1950. Furthermore, all three sites are ~300 m from a lake. Lakes disturb a profile if they are at distance less than the depth of the boreholes (Lewis and Wang, 1992). The Thierry Mine boreholes are 530 m or deeper. Therefore, they could be influenced by the presence of the lake. We hypothesize that the greater warming signal is related to the change in vegetation cover and the presence of the nearby lake (Lewis and Wang, 1992, 1998; Lewis, 1998).

6. *Lastly, one site (Eleonore) has warming that began considerably earlier than the other sites. Why might this be?*

We do not believe that Eleonore shows earlier warming. There is a clear recent warming signal.

7. *...Corvet, which is located on the side of a 30 m hill. However, what is the slope?*

*How much of an effect does this have? It is still being used, so the authors must think it isn't significant.*

Errors were made in the coordinates of Corvet in the manuscript. Corvet is located at $53°19.072'$ N and $73°55.760'$ W. Using the elevation of the Google Earth images of the site, we see that topography is less than 5 m. This will be fixed in the manuscript.

8. *The authors discuss what the LIA surface signal should be for the region, but do not see a ground signal. Perhaps a simple forward model of driving into the ground a surface temperature time series with the appropriate LIA signal and making a comparison to the boreholes would be appropriate? Then, the authors could argue whether the signal is strong enough to actually be observed, or whether it is not seen due to snow or something else. This is similar to the arm waving argument used to interpret a possible ground warming due to longer/deeper snow cover in the region, but it seems that other authors have performed some analysis that may provide quantitative support to their arguments (perhaps Bartlett et al., 2005?)*

The appropriate LIA signal for the region is unknown. Pollen has reconstructed a LIA signal of $\sim 0.3°C$ (Gajewski, 1988; Viau and Gajewski, 2009; Viau, 2012). Figure 1 shows than an LIA signal of less than 1 K in a noise free environment cannot be resolved with less than 5 singular values. As discussed above, noisy data requires larger cutoffs to decrease the impact of noise on the solution. This illustrates that a weak LIA signal cannot be resolved in a noisy environment.

9. *I did notice that on page 11 in the reference section that Jaupart and Mareschal,*

*2011 was published in Cambridge, not Cabridge; also, the next two references following the previous reference are of Jaupart et al., 2014 and are a duplicate.*

This will be fixed in the manuscript.

**References**

Gajewski, K.: Late Holocene climate changes in eastern North America estimated from pollen data, Quaternary Research, 29, 255–262, doi:10.1016/0033-5894(88)90034-8, 1988.

Jaupart, C., Mareschal, J., Bouquerel, H., and Phaneuf, C.: The building and stabilization of an Archean Craton in the Superior Province, Canada, from a heat flow perspective, Journal of Geophysical Research: Solid Earth, 119, 9130–9155, doi:10.1002/2014JB011018, 2014.

Lévy, F., Jaupart, C., Mareschal, J.-C., Bienfait, G., and Limare, A.: Low heat flux and large variations of lithospheric thickness in the Canadian Shield, Journal of Geophysical Research: Solid Earth (1978–2012), 115, doi:10.1029/2009JB006470, 2010.

Lewis, T.: The effect of deforestation on ground surface temperatures, Global and Planetary Change, 18, 1–13, doi:10.1016/S0921-8181(97)00011-8, 1998.

Lewis, T. J. and Wang, K.: Influence of terrain on bedrock temperatures, Palaeogeography, Palaeoclimatology, Palaeoecology, 98, 87–100, doi:10.1016/0031-0182(92)90190-G, 1992.

Lewis, T. J. and Wang, K.: Geothermal evidence for deforestation induced warming: Implications for the climatic impact of land development, Geophysical Research Letters, 25, 535–538, doi:10.1029/98GL00181, 1998.

Mareschal, J.-C. and Beltrami, H.: Evidence for recent warming from perturbed geothermal gradients: examples from eastern Canada, Climate Dynamics, 6, 135–143, doi:10.1007/BF00193525, 1992.

Viau, A. and Gajewski, K.: Reconstructing millennial-scale, regional paleoclimates of boreal Canada during the Holocene, Journal of Climate, 22, 316–330, doi:10.1175/2008JCLI2342.1, 2009.

Viau, A. E.: Climate of the last 2,000 years in North America recon-structed from pollen records, Quaternary International, 279, 520, doi:10.1016/j.quaint.2012.08.1800, 2012.

[Figure]

[Figure]

**Fig. 1.** GST reconstruction of a synthetic noise-free1 K cooling signal between 1600 and 1800 with varying singular values.

**Fig. 2.** Location of Thierry Mine boreholes on a satellite image.

---

## Referee Comment (RC2) · T. Alan (Referee) · 18 Oct 2016

Climate trends in northern Ontario and Quebec from borehole temperature profiles by C. Pickler, H. Beltrami, and J.-C. Mareschal

Reviewed by Alan E. Taylor, retired GSC

The authors have used boreholes of opportunity in northern Ontario and Quebec with temperature inversion techniques to reconstruct the ground surface temperature history for this area and hence an estimate of climate change over the past several centuries. The present paper is one of a long lineage of similar reconstructions from this group, employing a similar geophysical/mathematical methodology. It is well done.

My main suggestion is that at least a further brief description should be made to the

immediate borehole surface character and subsurface geology, even though this might be more fully covered in the referenced work on these boreholes. Indicate the predominant rock type, possibly with a profile of thermal conductivity that might explain some variations seen in the temperatures. Equations in sec. 2 indicate that conductivity profiles are part of the program input data with the temperature residuals. I assume there is not significant layering except at Nielson where the thermal depth is used to reduce the effect of conductivity variations.

The authors do mention other physical elements or processes that may affect the results (and conclusions?): snow cover or lack of, borehole area vegetation/forest cover and peat, drainage, surface overburden (any?). These might be augmented by the authors' visual observations of the immediate borehole surrounds during logging visits.

For Camp Coulon, the 3 temperature profiles do appear to be distinctly different (Fig. 3). The authors combine such closely spaced holes for an ensemble inversion, a usual practice. But one might wonder if separate inversions would give very different reconstructed temperature histories that might suggest other factors at play (thermal conductivity? Water flow?). Particularly the odd profile at CC1012, compared to nearby holes and considering such precise temperature measurements.

The authors acknowledge indications of subsurface water flows. Presumably (?) the holes are uncased. Could the consistent offsets in temperatures at N1015 ($\sim$300 m), N1013 ($\sim$240 m) and N1012 ($\sim\sim$160 m) possibly result from a sub-horizontal water flow (Fig. 2, Ontario) (e.g., geophysical models of Broedehoeft and others). These holes seem spatially very close (how much?) while N1012 is further south, so perhaps justifying the simultaneous inversion, but also questioning the reason for the differing temperature profiles. Similar offset feature at Eleonore ($\sim$350 m; Fig. 3, Quebec).

Permafrost. This section justifiably has raised issues with other commentators. At the southern margins of permafrost, predicting present or absence of persistent frozen ground is almost intractable even at the local scale: the experiences in Fairbanks,

Alaska bear witness (permafrost conferences). Local-scale, very near surface single point or shallow temperature cables time-series in the sporadic permafrost of the Mackenzie Valley show that surface vegetation, saturation, snow cover may reduce or eliminate the winter cooling cycle (e.g.,Taylor 2000 and similar results in Schefferville – Nicholson and Granberg's work); the topic is covered exhaustively in Zhang et al.'s (2005) work cited.

Maybe better here for the authors to conclude that their GST reconstructions show the potential for permafrost across the region is minimal to absent over the past 5 centuries, with its occurrence highly dependent on surface character and snow cover effects. Suggest add a citation for nearby locations, if any, where permafrost has been documented.

Small point: the temperature scale ranges vary through Fig. 2 and 3 but the sensitivities are consistent; it would be useful to indicate that in the caption as it makes for inter-comparison; same for Fig. 4-6. A few editorial suggestions are on an annotated copy of the PDF manuscript.

Please also note the supplement to this comment:
http://www.clim-past-discuss.net/cp-2016-55/cp-2016-55-RC2-supplement.pdf

**Supplement:**

[revised manuscript text omitted]

---

## Author Response (AR1)

Dear Dr. Winguth,

Please find our revised manuscript of "Climate trends in northern Ontario and Quebec from borehole temperature profiles" (cp-2016-55). We thank the anonymous reviewer and A.E.Taylor for their reviews. We appreciate the useful comments by R.Way. In response to their suggestions and questions, we have made several changes to the manuscript and added some information to clarify some points about data collection and processing. A detailed list and explanation of the changes follows. The changes have been highlighted on the attached copy of the revised manuscript.

**Response to Comments**

**In response to Anonymous Reviewer**

*1) Anonymous Reviewer: Borehole sites: In adequately determining if borehole sites are appropriate for use in climate reconstructions, several criteria are required. While the authors have addressed several of the sites and determined they were unsuitable (as presented in Table 2 of the manuscript), information regarding the other sites is not included that would aid a reader in understanding the conditions at the boreholes. For example, no discussion of slope, topography, vegetation or surface material is given, although the authors do reference previous studies. A discussion of vegetation and ground cover at the sites would be extremely useful, however, especially considering that the argument that one site (Thierry Mine) may have additional warming due to the removal of vegetation was put forth. Further, some sites are said to be "too shallow" or on*

*the side of steep hills. What exactly is too shallow and steep? Can a quantitative discussion replace the qualitative explanation? Also, are all of the boreholes vertical? At least one site was excluded because it was plunging under a lake. It should be clear.*

We agree with the reviewer. The dip of the boreholes varies between $40°$ and $90°$. This has been added to Table 1 (P21). Furthermore, a more detailed description of each site will be provided in an appendix.

In the original manuscript, we qualitatively explained why some boreholes were rejected (too shallow, proximity to a lake, slope, etc.). This means that holes less than 300 m were rejected for being too shallow. Holes were rejected for being near a lake when the mean distance was less than the depth of the hole, or less than 300 m. Holes were deemed too steep and rejected if they had slope of 5% or more over distance comparable to depth. This explanation has been added to the revised manuscript (P5L33, P6L1-2).

*2) Anonymous Reviewer: … if the results of the removal of the steady state gradient as shown in Figures 2 and 3 are different if the length (100 m) is modified?*

In their heat flow studies, Lévy et al. (2010) and Jaupart et al. (2014) have determined the heat flow for the majority of the boreholes presented in this manuscript and estimated how heat flux varies with the depth interval where it is estimated. They found that, below 300 m, the standard deviation on the estimates of the heat flux is less than 5% of the mean suggesting that the reference temperature profile does not vary much with depth. This has been clarified in the revised manuscript (P6L11-12).

*3) Anonymous Reviewer: The authors state that only one site has a ground surface temperature (GST) that was affected by the LIA. However, based on the temperature anomalies shown in Figures 2 and 3, it would seem that other sites exhibit cooling at the same depths as Otoskwin. Mussellwhite, TM0608, and CC0713 all have temperature anomalies that indicate cooling at the same general depth. Is this not a LIA signature*

The reviewer is correct that the temperature anomalies of Musselwhite, TM0608, and CC0713 indicate cooling. This has been clarified in the revised manuscript (P6L25, P7L2-3).

The profiles and anomalies at Musselwhite and TM0608 are noisy, which could mask a clear LIA signal when inverting the profiles. (Mareschal and Beltrami, 1992) showed that resolution decreases when noise and errors must be filtered. Large singular values are required when dealing with noisy data since they reduce the impact of noise but retain the gross features of the solution. Furthermore, a test was run with a 1 K cooling between 1600 and 1800 and varying the singular value cutoff. This showed that for noise-free synthetic data, a 1 K cooling cannot be resolved with less than 5 singular values. The CC0713 temperature anomaly is less noisy and shows a mild cooling of $\leq 0.2$ K. Our test shows that a cooling of this magnitude cannot be resolved in our reconstructions. This was further confirmed when we inverted the site individually and observed no LIA signal. This has been added in the revised manuscript (P7L29-

33, P8L1-2).

*4) Anonymous Reviewer: Also, the anomaly shown in CC0712 (Figure 3, top left) has a very interesting profile. What is the cause?*

The reviewer is correct. These discontinuities are also observed in the gradient between 100 and 300 m, which could be due to small water flows. This has been added to the revised manuscript (P7L3-5).

*5) Anonymous Reviewer: Other questions I have about the results that don't have any explanation or that aren't adequately explained include the assertion that the Thierry Mine signal may be amplified by the clearing of vegetation between 1934 and 1950. However, most of the GST histories show a large increase in temperature at this same time, indicating it may not be vegetation alone. Have the authors done any modeling or do they have any surface temperatures to support this hypothesis*

The warming at Thierry Mine is greater than the other sites (at least 0.7 K greater than any other site). During sampling, a large clearing near all three sites was noted. This was further confirmed by examining a satellite image of the region, which located the boreholes ∼500 m from a large clearing. In the original manuscript, we hypothesize this warming to be the result of vegetation clearing Lewis and Wang (1998); Lewis (1998).

However, the satellite image also locates the three boreholes ∼300 m

from a lake. Lakes disturb a profile if they are at distance less than the depth of the boreholes (Lewis and Wang, 1992). The Thierry Mine boreholes are 530 m or deeper. Therefore, they could be influenced by the presence of the lake. We hypothesize that the greater warming signal is related to the change in vegetation cover and the presence of the nearby lake (Lewis and Wang, 1992, 1998; Lewis, 1998). This has been added to and clarified in the revised manuscript (P7L13-14).

*6) Anonymous Reviewer: Lastly, one site (Eleonore) has warming that began considerably earlier than the other sites. Why might this be?*

We do not believe Eleonore shows earlier warming. There is a clear recent warming signal (Figure 6).

*7) Anonymous Reviewer: ...Corvet, which is located on the side of a 30 m hill. However, what is the slope? How much of an effect does this have? It is still being used, so the authors must think it isn't significant.*

Errors were made in the coordinates of Corvet in the manuscript. Corvet is located at $53°19.072'$N and $73°55.760'$W. Using the elevation of Google Earth images of the site, we see that the topography is less than 5m. This has been corrected in the manuscript.

*8) Anonymous Reviewer: The authors discuss what the LIA surface*

*signal should be for the region, but do not see a ground signal. Perhaps a simple forward model of driving into the ground a surface temperature time series with the appropriate LIA signal and making a comparison to the boreholes would be appropriate? Then, the authors could argue whether the signal is strong enough to actually be observed, or whether it is not seen due to snow or something else. This is similar to the arm waving argument used to interpret a possible ground warming due to longer/deeper snow cover in the region, but it seems that other authors have performed some analysis that may provide quantitative support to their arguments (perhaps Bartlett et al., 2005?)*

An appropriate LIA signal for the region is unknown. Pollen data have reconstructed a $\sim 0.3°$ LIA signal Gajewski (1988); Viau and Gajewski (2009); Viau (2012). We ran with a 1 K cooling between 1600 and 1800 and varying singular values. At least five singular values were required to resolve the signal. Furthermore, noisy data requires larger cutoffs to decrease the impact of noise on the solution (Mareschal and Beltrami, 1992). This illustrates that a weak LIA signal cannot be resolved in a noisy environment. This has been clarified in the revised manuscript (P7L29-34, P8L1-2).

*9) Anonymous Reviewer: I did notice that on page 11 in the reference section that Jaupart and Mareschal, 2011 was published in Cambridge, not Cabridge; also, the next two references following the previous reference are of Jaupart et al., 2014 and are a duplicate.*

This has been fixed in the revised manuscript.

**In response to comments by A.E.Taylor**

*10) A.E.Taylor: My main suggestion is that at least a further brief description should be made to the immediate borehole surface character and subsurface geology, even though this might be more fully covered in the referenced work on these boreholes. Indicate the predominant rock type, possibly with a profile of thermal conductivity that might explain some variations seen in the temperatures.*

We agree with the reviewer that a more detailed description of each site is useful. We have added the dip of the boreholes to Table 1 (P21). Furthermore, a detailed description of each site will be provided in an appendix.

*11) A.E.Taylor: Equations in sec. 2 indicate that conductivity profiles are part of the program input data with the temperature residuals. I assume there is not significant layering except at Nielson where the thermal depth is used to reduce the effect of conductivity variations.*

The reviewer is correct. Nielsen Island is the only site to present significant layering, which is accounted for by using the thermal depth to account for the conductivity variations. This is noted on P5, L31-32 of the revised manuscript.

*12) A.E.Taylor: The authors do mention other physical elements or*

*processes that may affect the results (and conclusions?): snow cover or lack of, borehole area vegetation/forest cover and peat, drainage, surface overburden (any?). These might be augmented by the authors visual observations of the immediate borehole surrounds during logging visits.*

Visual observations of the immediate borehole surroundings are routinely noted in our log books when logging the boreholes. Furthermore, we examine the borehole locations on satellite images and topographic maps. This will be included in an appendix with a detailed description of each site.

*13) A.E.Taylor: For Camp Coulon, the 3 temperature profiles do appear to be distinctly different (Fig. 3). The authors combine such closely spaced holes for an ensemble inversion, a usual practice. But one might wonder if separate inversions would give very different reconstructed temperature histories that might suggest other factors at play (thermal conductivity? Water flow?). Particularly the odd profile at CC1012, compared to nearby holes and considering such precise temperature measurements.*

The three Camp Coulon sites (0712, 0713, 0714) were inverted individually. Similar trends are observed throughout the three GST reconstructions. A warming signal of ∼1-1.5 K occurs at ∼100 years BP. The maximum temperature at 0712, however, is reached 20 years before that of 0713 and 0714. Differences between the reconstructions could be the result of noise in the profiles. Therefore, we chose to simultaneously invert the profiles to increase the signal to noise ratio.

*14) ...Presumably (?) the holes are uncased.*

The holes are cased for the upper ∼10-15 m, usually until bedrock is reached. This has been clarified in the revised manuscript (P5L17-18).

*15) Could the consistent offsets in temperatures at N1015 (∼300 m), N1013 (∼240 m), and N1012 (∼160 m) possibly result from a sub-horizontal water flow (Fig. 2, Ontario) (e.g., geophysical models of Broedehoeft and others). These holes seem spatially very close (how much?) while N1012 is further south, so perhaps justifying the simultaneous inversion, but also questioning the reason for the differing temperature profiles. Similar offset feature at Eleonore (∼350 m, Fig. 3, Quebec).*

The reviewer is correct. The consistent offsets in temperatures at N1015, N1013, and N1012 could be the result of a sub-horizontal water flow. We examined our log book for any signs of oscillation or instability during the temperature measurements, often a sign of water flow. Several unstable measurements were noted at N1013. However, no instability was noted at the other two sites (N1012 and N1015) but the temperature gradient did reverse. This could be attributed to water flow.

The holes are spatially very close. There was an error made in the coordinates of N1015, which are $52°44'25.5''$N and $86°18'11.9''$W. This has been fixed in the revised manuscript (Table 1, P21).

*16) Maybe better here for the authors to conclude that their GST reconstructions show the potential for permafrost across the region is minimal to absent over the past 5 centuries, with its occurrence highly dependent on surface character and snow cover effects. Suggest add a citation for nearby locations, if any, where permafrost has been documented.*

We agree with the reviewer. We have better qualified our conclusions by stating that the GST reconstructions show a minimal potential for permafrost across the region for the past five centuries in the revised manuscript (P1L7-8, P8L32-33, P9L1).

Thibault and Payette (2009) document permafrost in the James Bay peatland bogs near the Northern Quebec sites of Lagrande, Eleonore, and Eastmain. This citation has been added to the manuscript (P8L29-30).

*17) ...the temperature scale ranges vary through Fig. 2 and 3 but the sensitivities are consistent; it would be useful to indicate that in the caption as it makes for inter-comparison; same for Fig. 4-6. A few editorial suggestions are on an annotated copy of the PDF manuscript*

We appreciated the suggestions and have incorporated them into the manuscript.

We have also considered the comments from yourself and R.Way when revising the manuscript.

**In response to R.Way**

*16) The authors incorrectly state that their sample sites are all located within the sporadic (10-50% of the land surface) to extensive discontinuous zones (50-90% of the land surface). In nearly every case the sampled sites are located within the isolated patches permafrost zone (<10% of land surface) according to maps produced by Heginbottom et al (1995), Payette (2001) and recent spatial numerical modelling of permafrost distribution for Labrador-Ungava (Way and Lewkowicz, 2016). Considering the more realistic permafrost extent, there is no discrepancy between the borehole observations and existing permafrost maps.*

We are thankful to R.Way for pointing out that the map we were using is out of date. According to the map from the geological survey, our boreholes lie in a region of discontinuous permafrost. In Quebec, we cover a region classified as having only isolated patches of permafrost. In northernOntario, Noront (4 holes) is near the southern edge of a region with extensive discontinuous permafrost (50-90%). This has been fixed in the revised manuscript (P3L9-14,P5L26-30, P8L34).

For regions of discontinuous isolated patches of permafrost, where we did not find permafrost at any site, our statistics are not sufficient to draw any conclusion. However, for Noront, near the southern boundary of extensive discontinuous permafrost, using the low frequency of 50%, the probability that not one of 4 holes meets permafrost is low: 1/16 or 6.25%. Furthermore, in Manitoba, we had sampled a region of sporadic discontinuous permafrost (10- 50%) between the towns of Flin Flon, Thompson, and Lynn Lake, but we found that only 1 hole in more than 60 was blocked, as described and discussed by Guillou-Frottier et al. (1998). Using the lowest value of 10% permafrost coverage, the probability that only one hole out of 60 encounters permafrost is only 0.1 x $10^{-59}$ x 60 or 1.20%; for the 50% frequency, the probability is only 5.2 x $10^{-17}$. This demonstrates a possible discrepancy between borehole observations and existing permafrost maps. We do, however, believe that we must better qualify our conclusions by stating that our reconstructions indicate that a minimal potential for permafrost across the region for the past five centuries. This has been clarified in the manuscript (P1L7-8, P8L32-33, P9L1).

*17) R.Way: A further point on a similar subject is that the temperature sampling methodology here is at too coarse a resolution (depth) to detect thin permafrost bodies if they were to exist. In the southern end of the discontinuous zone you would be more likely to find thinner permafrost bodies therefore this is a serious limitation of the study.*

We disagree. To avoid repetition of the measurement procedure, we did not sufficiently emphasize the obvious fact that the temperature measurements in boreholes must be performed a long time after drilling when the ground has returned to thermal equilibrium. Should permafrost be present, the hole would be frozen and we would not be able to lower the probe in it to measure temperature. There is no permafrost in the holes that we measured regardless of

the depth sampling interval. A second point is that thin permafrost bodies are very unlikely to be present for simple reasons of thermodynamics. The second law of thermodynamics implies that any temperature oscillation decays with time, and it follows that in steady state the vertical temperature profile can have no maximum or minimum, implying that the existence of a thin frozen layer is impossible in steady state. A frozen underground layer could exist but only as a transient. The life time of such a layer depends on its thickness. For standard values of thermal diffusivity, a 1m thick layer would last less than 1 year.

*18) The article also does not present any indication of the land cover types encountered in the study area and correspondingly, does not consider how permafrost is distributed across the landscape (e.g. Shur and Jorgenson, 2007; Jorgenson et al., 2010). In northern Ontario and Quebec, permafrost is largely absent from forested areas at the southern end of the discontinuous zones where snow accumulates while concurrently being present on wind exposed mountaintops (Brown, 1979; Ives, 1979; Allard and Séguin, 1987; Granberg, 1989; Ou et al. 2016a,b; Way and Lewkowicz, 2016). Ignoring these critical variables makes it untenable to draw large-scale conclusions on permafrost from the provided data.*

R.Way is correct that we did not consider the landscape in these studies. The main reason being that we use holes of opportunity that were drilled for mining exploration in any surface environment. They represent thus a random unbiased sampling of the landscape. Outside Quebec, the study area is in lowlands without "mountaintops". The topography near our sites on the Quebec side is more

marked but remains low. Most likely, disagreements arise from the inconsistent and poor quality of the permafrost data because of insufficient sampling and the extreme spatial variability of the land surface.

*19) In general, I believe that the discussion of permafrost in this article should be removed in its entirety as the methodology, discussion and interpretations presented are not appropriate for the analysis of permafrost distribution and history. Finally, the lack of consideration of the literature on permafrost in western Quebec and northern Ontario must be addressed.*

We appreciate the comment but must disagree. The methodology is sound, and our reconstruction of past temperature histories is based on very simple physics. As much as we understand the need for considering the literature on permafrost, we believe that permafrost studies would gain to be confronted with physical models. We do, however, believe that we must better qualify our conclusions by stating that our reconstructions indicate that a minimal potential for permafrost across the region for the past five centuries. This has been done in the revised manuscript (P1L7-8, P8L32-33, P9L1).

**In response to Editor: A.Winguth**

*20) I noticed that the temperature change ("1-2 K") and the time period ("for the last 150 years") in the abstract and discussion and conclusion are vague. Please replace the temperature range with mean and standard deviation and specify the time period (e.g "for*

*the period 1850 to 2000").*

This has been fixed in the revised manuscript (P1L4, P7,L10-11, P8L25).

We trust that we have addressed all the comments and that our revisions have resulted in improving significantly the manuscript.

Sincerely yours,

Carolyne Pickler
Hugo Beltrami
Jean-Claude Mareschal

**References**

[revised manuscript text omitted]
 | 0603 | $51^\circ 34' 51''$ | $88^\circ 31' 09''$ | 3.33$\pm$0.02 | 25.8$\pm$1.5 | Too shallow | (Lévy et al., 2010) |
| LaGrande | 0406 | $53^\circ 31' 42''$ | $76^\circ 33' 49''$ | 2.67$\pm$0.005 | 14.1$\pm$0.3 | Topography | (Lévy et al., 2010) |
| Eleonore | 0503 | $52^\circ 42' 00''$ | $76^\circ 04' 45''$ | 2.56$\pm$0.01 | 27.7$\pm$0.6 | Reservoir | (Lévy et al., 2010) |
| Clearwater | 0505 | $52^\circ 12' 33''$ | $75^\circ 48' 38''$ | 2.33$\pm$0.02 | 30.6$\pm$0.4 | Lake | (Lévy et al., 2010) |
| | 0506 | $52^\circ 12' 31''$ | $75^\circ 48' 23''$ | 2.23$\pm$0.01 | 31.3$\pm$0.2 | Lake | (Lévy et al., 2010) |
| | 0507 | $52^\circ 12' 39''$ | $75^\circ 48' 23''$ | 2.73$\pm$0.004 | 30.5$\pm$0.3 | Lake | (Lévy et al., 2010) |
| Poste Lemoyne | 0715 | $53^\circ 27' 37''$ | $75^\circ 12' 21''$ | 1.83$\pm$0.004 | 24.5$\pm$0.2 | Topography | (Lévy et al., 2010) |

**Table 3.** Summary GST History Results where $T_o$ is the reference surface temperature, $Q_o$ is the reference heat flux, $\Delta T$ is the difference between the maximal temperature and the reference temperature 500 yrs before logging.

| Site | Log ID | year | $T_o$ (°C) | $Q_o$ (mW m$^{-2}$) | $\Delta T$ (K) | LIA |
|------|--------|------|------|------|------|-----|
| Musselwhite | 0601 | 2006 | 2.56±0.01 | 30.0±0.5 | 0.88 | no |
| Thierry Mine | | | | | 2.85 | no |
| | 0605 | 2006 | 2.63±0.01 | 25.9±0.5 | | |
| | 0606 | 2006 | 2.55±0.01 | 23.5±0.7 | | |
| | 0608 | 2006 | 2.62±0.01 | 23.9±0.3 | | |
| Otoskwin | n/a | 1985 | 2.81±0.001 | 19.5±0.05 | 0.50 | yes |
| Noront | | | | | 1.85 | no |
| | 1012 | 2010 | 1.16±0.01 | 35.5±0.9 | | |
| | 1013 | 2010 | 1.51±0.02 | 31.3±1.5 | | |
| | 1014 | 2010 | 1.78±0.004 | 32.2±0.3 | | |
| | 1015 | 2010 | 2.35±0.02 | 27.6±1.4 | | |
| Nielsen Island | 0-77 | 1977 | -0.43±0.02 | 23.7±0.4 | 1.35 | no |
| LaGrande | 0405 | 2004 | 2.18±0.004 | 19.4±0.2 | 0.50 | no |
| Eastmain | | | | | 2.15 | no |
| | 0803 | 2008 | 2.98±0.01 | 27.3±0.4 | | |
| | 0804 | 2008 | 3.01±0.003 | 27.0±0.2 | | |
| Eleonore | 0502 | 2005 | 2.04±0.01 | 30.9±0.5 | 1.46 | no |
| Corvet | 0716 | 2007 | 2.45±0.01 | 24.9±0.3 | 2.18 | no |
| Camp Coulon | | | | | 1.34 | no |
| | 0712 | 2007 | 1.46±0.01 | 26.6±0.4 | | |
| | 0713 | 2007 | 1.83±0.01 | 24.2±0.6 | | |
| | 0714 | 2007 | 1.33±0.004 | - | | |

**Table A1.** Geological unit and rock type concerning the boreholes used in this study

| Site | Log ID | Rock Type | Geological Unit | Reference |
|------|--------|-----------|-----------------|-----------|
| Musselwhite | 0601 | gneiss | Sachigo subprovince | Lévy et al. (2010) |
| Thierry Mine | 0605 | granite | Uchi belt | Lévy et al. (2010) |
| | 0606 | | | Lévy et al. (2010) |
| | 0608 | | | Lévy et al. (2010) |
| Otoskwin | n/a | - | - | Jessop and Lewis (1978) |
| Noront | 1012 | - | North Caribou terrane | Jaupart et al. (2014) |
| | 1013 | | | Jaupart et al. (2014) |
| | 1014 | | | Jaupart et al. (2014) |
| | 1015 | | | Jaupart et al. (2014) |
| Nielsen Island | 0-77 | - | - | Jessop (1968) |
| LaGrande | 0405 | granodiorite | LaGrande volcano-plutonic belt | Lévy et al. (2010) |
| Eastmain | 0803 | metasedimentary and volcanics | LaGrande volcano-plutonic belt | Jaupart et al. (2014) |
| | 0804 | | | Jaupart et al. (2014) |
| Eleonore | 0502 | wacke | LaGrande volcano-plutonic belt | Lévy et al. (2010) |
| Corvet | 0716 | intermediate volcanics | LaGrande volcano-plutonic belt | Lévy et al. (2010) |
| Camp Coulon | 0712 | rhyolite | LaGrande volcano-plutonic belt | Lévy et al. (2010) |
| | 0713 | | | Lévy et al. (2010) |
| | 0714 | | | Lévy et al. (2010) |